# What the Vec?
# Towards Probabilistically Grounded Embeddings

**Carl Allen**[1]  **Ivana Balažević**[1]  **Timothy Hospedales**[1,2]
[1] School of Informatics, University of Edinburgh, UK
[2] Samsung AI Centre, Cambridge, UK
{carl.allen, ivana.balazevic, t.hospedales}@ed.ac.uk

## Abstract

Word2Vec (W2V) and GloVe are popular, fast and efficient word embedding algorithms. Their embeddings are widely used and perform well on a variety of natural language processing tasks. Moreover, W2V has recently been adopted in the field of graph embedding, where it underpins several leading algorithms. However, despite their ubiquity and relatively simple model architecture, a theoretical understanding of *what* the embedding parameters of W2V and GloVe learn and *why* that is useful in downstream tasks has been lacking. We show that different interactions between *PMI vectors* reflect semantic word relationships, such as similarity and paraphrasing, that are encoded in low dimensional word embeddings under a suitable projection, theoretically explaining why embeddings of W2V and GloVe work. As a consequence, we also reveal an interesting mathematical interconnection between the considered semantic relationships themselves.

## 1   Introduction

Word2Vec[1] (W2V) [25] and GloVe [29] are fast, straightforward algorithms for generating *word embeddings*, or vector representations of words, often considered points in a *semantic space*. Their embeddings perform well on downstream tasks, such as identifying word similarity by vector comparison (e.g. cosine similarity) and solving analogies, such as the well known "*man* is to *king* as *woman* is to *queen*", by the addition and subtraction of respective embeddings [26, 27, 19].

In addition, the W2V algorithm has recently been adopted within the growing field of *graph embedding*, where the typical aim is to represent graph nodes in a common latent space such that their relative positioning can be used to predict edge relationships. Several state-of-the-art models for graph representation incorporate the W2V algorithm to learn node embeddings based on random walks over the graph [13, 30, 31]. Furthermore, word embeddings often underpin embeddings of word sequences, e.g. sentences. Although sentence embedding models can be complex [8, 17], as shown recently [38] they sometimes learn little beyond the information available in word embeddings.

Despite their relative ubiquity, much remains unknown of the W2V and GloVe algorithms, perhaps most fundamentally we lack a theoretical understanding of (i) *what is learned* in the embedding parameters; and (ii) *why that is useful* in downstream tasks. Answering such core questions is of interest in itself, particularly since the algorithms are unsupervised, but may also lead to improved embedding algorithms, or enable better use to be made of the embeddings we have. For example, both algorithms generate two embedding matrices, but little is known of how they relate or should interact. Typically one is simply discarded, whereas empirically their mean can perform well [29] and elsewhere they are assumed identical [14, 4]. As for embedding interactions, a variety of heuristics are in common use, e.g. cosine similarity [26] and *3CosMult* [19].

Of works that seek to theoretically explain these embedding models [20, 14, 4, 9, 18], Levy and Goldberg [20] identify the loss function minimised (implicitly) by W2V and, thereby, the relationship between W2V word embeddings and the *Pointwise Mutual Information* (PMI) of word co-occurrences. More recently, Allen and Hospedales [2] showed that this relationship explains the linear interaction observed between embeddings of analogies. Building on these results, our key contributions are:

- to show how particular semantic relationships correspond to linear interactions of high dimensional *PMI vectors* and thus to equivalent interactions of low dimensional word embeddings generated by their *linear* projection, thereby explaining the semantic properties exhibited by embeddings of W2V and GloVe;

- to derive a relationship between embedding matrices proving that they must differ, justifying the heuristic use of their mean and enabling word embedding interactions – including the widely used cosine similarity – to be semantically interpreted; and

- to establish a novel hierarchical mathematical inter-relationship between relatedness, similarity, paraphrase and analogy (Fig 2).

## 2   Background

**Word2Vec** [25, 26] takes as input word pairs $\{(w_{i_r}, c_{j_r})\}_{r=1}^D$ extracted from a large text corpus, where target word $w_i \in \mathcal{E}$ ranges over the corpus and context word $c_j \in \mathcal{E}$ ranges over a window of size $l$, symmetric about $w_i$ ($\mathcal{E}$ is the dictionary of distinct words, $n = |\mathcal{E}|$). For each observed word pair, $k$ random pairs (*negative samples*) are generated from unigram distributions. For embedding dimension $d$, W2V's architecture comprises the product of two weight matrices $\mathbf{W}, \mathbf{C} \in \mathbb{R}^{d \times n}$ subject to the logistic sigmoid function. Columns of $\mathbf{W}$ and $\mathbf{C}$ are the *word embeddings*: $\mathbf{w}_i \in \mathbb{R}^d$, the $i^{\text{th}}$ column of $\mathbf{W}$, represents the $i^{th}$ word in $\mathcal{E}$ observed as the target word ($w_i$); and $\mathbf{c}_j \in \mathbb{R}^d$, the $j^{\text{th}}$ column of $\mathbf{C}$, represents the $j^{th}$ word in $\mathcal{E}$ observed as a context word ($c_j$).

Levy and Goldberg [20] show that the loss function of W2V is given by:

$$\ell_{W2V} = -\sum_{i=1}^{n}\sum_{j=1}^{n} \#(w_i, c_j) \log \sigma(\mathbf{w}_i^\top \mathbf{c}_j) + \tfrac{k}{D}\#(w_i)\#(c_j) \log(\sigma(-\mathbf{w}_i^\top \mathbf{c}_j)), \tag{1}$$

which is minimised if $\mathbf{w}_i^\top \mathbf{c}_j = \mathbf{P}_{i,j} - \log k$, where $\mathbf{P}_{i,j} = \log \frac{p(w_i, c_j)}{p(w_i)p(c_j)}$ is *pointwise mutual information* (PMI). In matrix form, this equates to factorising a *shifted* PMI matrix $\mathbf{S} \in \mathbb{R}^{n \times n}$:

$$\mathbf{W}^\top \mathbf{C} = \mathbf{S} . \tag{2}$$

**GloVe** [29] has the same architecture as W2V, but a different loss function, minimised when:

$$\mathbf{w}_i^\top \mathbf{c}_j = \log p(w_i, c_j) - b_i - b_j + \log Z, \tag{3}$$

for biases $b_i$, $b_j$ and normalising constant $Z$. In principle, the biases provide flexibility, broadening the family of statistical relationships that GloVe embeddings can learn.

**Analogies** are word relationships, such as the canonical "*man* is to *king* as *woman* is to *queen*", that are of particular interest because their word embeddings appear to satisfy a linear relationship [27, 19]. Allen and Hospedales [2] recently showed that this phenomenon follows from relationships between *PMI vectors*, i.e. rows of the (unshifted) PMI matrix $\mathbf{P} \in \mathbb{R}^{n \times n}$. In doing so, the authors define (i) the *induced distribution* of an observation ∘ as $p(\mathcal{E}|\circ)$, the probability distribution over all context words observed given ∘; and (ii) that a word $w_*$ *paraphrases* a set of words $\mathcal{W} \subset \mathcal{E}$ if the induced distributions $p(\mathcal{E}|w_*)$ and $p(\mathcal{E}|\mathcal{W})$ are (elementwise) similar.

## 3   Related Work

While many works explore empirical properties of word embeddings (e.g. [19, 23, 5]), we focus here on those that seek to theoretically explain why W2V and GloVe word embeddings capture semantic properties useful in downstream tasks. The first of these is the previously mentioned derivation by Levy and Goldberg [20] of the loss function (1) and the PMI relationship that minimises it (2). Hashimoto et al. [14] and Arora et al. [4] propose generative language models to explain the structure

found in word embeddings. However, both contain strong *a priori* assumptions of an underlying geometry that we do not require (further, we find that several assumptions of [4] fail in practice (Appendix D)). Cotterell et al. [9] and Landgraf and Bellay [18] show that W2V performs *exponential (binomial) PCA* [7], however this follows from the (binomial) negative sampling and so describes the algorithm's mechanics, not *why* it works. Several works focus on the linearity of analogy embeddings [4, 12, 2, 10], but only [2] rigorously links semantics to embedding geometry (S.2).

To our knowledge, no previous work explains how the semantic properties of relatedness, similarity, paraphrase and analogy are all encoded in the relationships of PMI vectors and thereby manifest in the low dimensional word embeddings of W2V and GloVe.

## 4 PMI: linking geometry to semantics

The derivative of W2V's loss function (1) with respect to embedding $\mathbf{w}_i$, is given by:

$$\frac{1}{D}\nabla_{\mathbf{w}_i}\ell_{W2V} = \sum_{j=1}^{n} \big(\underbrace{p(w_i,c_j) + kp(w_i)p(c_j)}_{\mathbf{d}_j^{(i)}}\big)\big(\underbrace{\sigma(\mathbf{S}_{i,j}) - \sigma(\mathbf{w}_i^\top\mathbf{c}_j)}_{\mathbf{e}_j^{(i)}}\big)\mathbf{c}_j = \mathbf{C}\,\mathbf{D}^{(i)}\mathbf{e}^{(i)}, \quad (4)$$

for diagonal matrix $\mathbf{D}^{(i)} = diag(\mathbf{d}^{(i)}) \in \mathbb{R}^{n\times n}$; $\mathbf{d}^{(i)}, \mathbf{e}^{(i)} \in \mathbb{R}^n$ containing the probability and error terms indicated; and all probabilities estimated empirically from the corpus. This confirms that (1) is minimised if $\mathbf{W}^\top\mathbf{C} = \mathbf{S}$ (2), since all $\mathbf{e}_j^{(i)} = 0$, but that requires $\mathbf{W}$ and $\mathbf{C}$ to each have rank at least that of $\mathbf{S}$. In the general case, including the typical case $d \ll n$, (1) is minimised when probability weighted error vectors $\mathbf{D}^{(i)}\mathbf{e}^{(i)}$ are orthogonal to the rows of $\mathbf{C}$. As such, embeddings $\mathbf{w}_i$ can be seen as a non-linear (due to the sigmoid function $\sigma(\cdot)$) *projection* of rows of $\mathbf{S}$, induced by the loss function. (Note that the distinction between $\mathbf{W}$ and $\mathbf{C}$ is arbitrary: embeddings $\mathbf{c}_j$ can also be viewed as projections onto the rows of $\mathbf{W}$.)

Recognising that the $\log k$ shift term is an artefact of the W2V algorithm (see Appendix A), whose effect can be evaluated subsequently (as in [2]), we exclude it and analyse properties and interactions of word embeddings $\mathbf{w}_i$ that are projections of $\mathbf{p}^i$, the corresponding rows of $\mathbf{P}$ (*PMI vectors*). We aim to identify the properties of PMI vectors that capture semantics and are then preserved in word embeddings under the low-rank projection induced by a suitably chosen loss function.

### 4.1 The domain of PMI vectors

PMI vector $\mathbf{p}^i \in \mathbb{R}^n$ has a component $\text{PMI}(w_i, c_j)$ for all context words $c_j \in \mathcal{E}$, given by:

$$\text{PMI}(w_i, c_j) = \log\frac{p(c_j, w_i)}{p(w_i)p(c_j)} = \log\frac{p(c_j|w_i)}{p(c_j)}. \quad (5)$$

Any difference in the probability of observing $c_j$ having observed $w_i$, relative to its marginal probability, can be thought of as *due to* $w_i$. Thus $\text{PMI}(w_i, c_j)$ captures the influence of one word on another. Specifically, by reference to marginal probability $p(c_j)$: $\text{PMI}(w_i, c_j) > 0$ implies $c_j$ is more likely to occur in the presence of $w_i$; $\text{PMI}(w_i, c_j) < 0$ implies $c_j$ is less likely to occur given $w_i$; and $\text{PMI}(w_i, c_j) = 0$ indicates that $w_i$ and $c_j$ occur independently, i.e. they are unrelated. PMI thus reflects the semantic property of *relatedness*, as previously noted [36, 6, 15]. A PMI *vector* thus reflects any change in the probability distribution over all words $p(\mathcal{E})$, given (or due to) $w_i$:

$$\mathbf{p}^i \triangleq \big\{\log\frac{p(c_j|w_i)}{p(c_j)}\big\}_{c_j \in \mathcal{E}} \triangleq \log\frac{p(\mathcal{E}|w_i)}{p(\mathcal{E})}. \quad (6)$$

While PMI values are unconstrained in $\mathbb{R}$, PMI vectors are constrained to an $n-1$ dimensional surface $\mathcal{S} \subset \mathbb{R}^n$, where each dimension corresponds to a word (Fig 1) (although technically a *hypersurface*, we refer to $\mathcal{S}$ simply as a "surface"). The geometry of $\mathcal{S}$ can be constructed step-wise from (6):

- the vector of numerator terms $\mathbf{q}^i = p(\mathcal{E}|w_i)$ lies on the simplex $\mathcal{Q} \subset \mathbb{R}^n$;
- dividing all $\mathbf{q} \in \mathcal{Q}$ (element-wise) by $\mathbf{p} = p(\mathcal{E}) \in \mathcal{Q}$, gives probability ratio vectors $\frac{\mathbf{q}}{\mathbf{p}}$ that lie on a "stretched simplex" $\mathcal{R} \subset \mathbb{R}^n$ (containing $\mathbf{1} \in \mathbb{R}^n$) that has a vertex at $\frac{1}{p(c_j)}$ on axis $j$, $\forall c_j \in \mathcal{E}$; and
- the natural logarithm transforms $\mathcal{R}$ to the surface $\mathcal{S}$, with $\mathbf{p}^i = \log\frac{p(\mathcal{E}|w_i)}{p(\mathcal{E})} \in \mathcal{S}$, $\forall w_i \in \mathcal{E}$.

Note, $\mathbf{p} = p(\mathcal{E})$ uniquely determines $\mathcal{S}$. Considering each point $\mathbf{s} \in \mathcal{S}$ as an element-wise log probability ratio vector $\mathbf{s} = \log \frac{\mathbf{q}}{\mathbf{p}} \in \mathcal{S}$ ($\mathbf{q} \in \mathcal{Q}$), shows $\mathcal{S}$ to have the properties (proofs in Appendix B):

**P1** $\mathcal{S}$**, and any subsurface of** $\mathcal{S}$**, is non-linear.** PMI vectors are thus not constrained to a linear subspace, identifiable by low-rank factorisation of the PMI matrix, as may seem suggested by (2).

**P2** $\mathcal{S}$ **contains the origin,** which can be considered the PMI vector of the *null word* $\emptyset$, i.e. $\mathbf{p}^\emptyset = \log \frac{p(\mathcal{E}|\emptyset)}{p(\mathcal{E})} = \log \frac{p(\mathcal{E})}{p(\mathcal{E})} = \mathbf{0} \in \mathbb{R}^n$.

**P3** **Probability vector** $\mathbf{q} \in \mathcal{Q}$ **is normal to the tangent plane of** $\mathcal{S}$ at $\mathbf{s} = \log \frac{\mathbf{q}}{\mathbf{p}} \in \mathcal{S}$.

**P4** $\mathcal{S}$ **does not intersect with the fully positive or fully negative orthants** (excluding $\mathbf{0}$). Thus PMI vectors are not *isotropically* (i.e. uniformly) distributed in space (as assumed in [4]).

**P5** **The sum of 2 points** $\mathbf{s} + \mathbf{s}'$ **lies in** $\mathcal{S}$ **only for certain** $\mathbf{s}, \mathbf{s}' \in \mathcal{S}$**.** That is, for any $\mathbf{s} \in \mathcal{S}$ ($\mathbf{s} \neq \mathbf{0}$), there exists a (strict) subset $\mathcal{S}_s \subset \mathcal{S}$, such that $\mathbf{s} + \mathbf{s}' \in \mathcal{S}$ *iff* $\mathbf{s}' \in \mathcal{S}_s$. Trivially $\mathbf{0} \in \mathcal{S}_s$, $\forall \mathbf{s} \in \mathcal{S}$.

Note that while all PMI vectors lie in $\mathcal{S}$, certainly not all (infinite) points in $\mathcal{S}$ correspond to the (finite) PMI vectors of words. Interestingly, P2 and P5 allude to properties of a *vector space*, often the desired structure for a *semantic space* [14]. Whilst the domain of PMI vectors is clearly not a vector space, addition and subtraction of PMI vectors do have *semantic meaning*, as we now show.

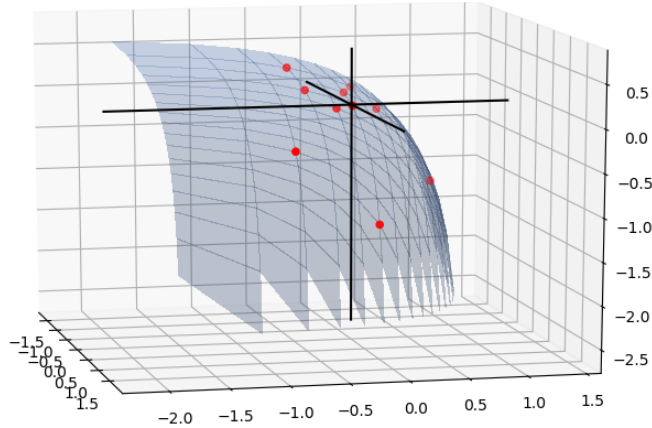

Figure 1: The PMI surface $\mathcal{S}$, showing sample PMI vectors of words (red dots)

## 4.2 Subtraction of PMI vectors finds similarity

Taking the definition from [2] (see S.2), we consider a word $w_i$ that *paraphrases* a word set $\mathcal{W} \in \mathcal{E}$, where $\mathcal{W} = \{w_j\}$ contains a single word. Since paraphrasing requires distributions of local context words (induced distributions) to be similar, this intuitively finds $w_i$ that are interchangeable with, or *similar* to, $w_j$: in the limit $w_j$ itself or, less trivially, a synonym. Thus, word similarity corresponds to a low KL divergence between $p(\mathcal{E}|w_i)$ and $p(\mathcal{E}|w_j)$. Interestingly, the difference between the associated PMI vectors:

$$\boldsymbol{\rho}^{i,j} = \mathbf{p}^i - \mathbf{p}^j = \log \frac{p(\mathcal{E}|w_i)}{p(\mathcal{E}|w_j)}, \tag{7}$$

is a vector of un-weighted KL divergence components. Thus, if dimensions were suitably weighted, the sum of difference components (comparable to Manhattan distance but *directed*) would equate to a KL divergence between induced distributions. That is, if $\mathbf{q}^i = p(\mathcal{E}|w_i)$, then a KL divergence is given by $\mathbf{q}^{i\top} \boldsymbol{\rho}^{i,j}$. Furthermore, $\mathbf{q}^i$ is the normal to the surface $\mathcal{S}$ at $\mathbf{p}^i$ (with unit $l_1$ norm), by P3. The projection onto the normal (to $\mathcal{S}$) at $\mathbf{p}^j$, i.e. $-\mathbf{q}^{j\top} \boldsymbol{\rho}^{i,j}$, gives the other KL divergence. (Intuition for the semantic interpretation of each KL divergence is discussed in Appendix A of [2].)

### 4.3 Addition of PMI vectors finds paraphrases

From geometric arguments (P5), we know that only certain pairs of points in $\mathcal{S}$ sum to another point in the surface. We can also consider the *probabilistic* conditions for PMI vectors to sum to another:

$$\mathbf{x} = \mathbf{p}^i + \mathbf{p}^j = \log \frac{p(\mathcal{E}|w_i)}{p(\mathcal{E})} + \log \frac{p(\mathcal{E}|w_j)}{p(\mathcal{E})}$$

$$= \underbrace{\log \frac{p(\mathcal{E}|w_i,w_j)}{p(\mathcal{E})}}_{\mathbf{p}^{i,j}} - \underbrace{\log \frac{p(w_i,w_j|\mathcal{E})}{p(w_i|\mathcal{E})p(w_j|\mathcal{E})}}_{\boldsymbol{\sigma}^{ij}} + \underbrace{\log \frac{p(w_i,w_j)}{p(w_i)p(w_j)}}_{\tau^{ij}} = \mathbf{p}^{i,j} - \boldsymbol{\sigma}^{ij} + \tau^{ij}\mathbf{1}, \quad (8)$$

where (overloading notation) $\mathbf{p}^{i,j} \in \mathcal{S}$ is a vector of PMI terms involving $p(\mathcal{E}|w_i, w_j)$, the induced distribution of $w_i$ and $w_j$ observed *together*;[2] and $\boldsymbol{\sigma}^{ij} \in \mathbb{R}^n$, $\tau^{ij} \in \mathbb{R}$ are the conditional and marginal dependence terms indicated (as seen in [2]). From (8), if $w_i$, $w_j \in \mathcal{E}$ occur both *independently and conditionally independently* given each and every word in $\mathcal{E}$, then $\mathbf{x} = \mathbf{p}^{i,j} \in \mathcal{S}$, and (from P5) $\mathbf{p}^j \in \mathcal{S}_{\mathbf{p}^i}$ and $\mathbf{p}^i \in \mathcal{S}_{\mathbf{p}^j}$. If not, error vector $\boldsymbol{\varepsilon}^{ij} = \boldsymbol{\sigma}^{ij} - \tau^{ij}\mathbf{1}$ separates $\mathbf{x}$ and $\mathbf{p}^{i,j}$ and $\mathbf{x} \notin \mathcal{S}$, unless by meaningless coincidence. (Note, whilst probabilistic aspects here mirror those of [2], we combine these with a geometric understanding.) Although certainly $\mathbf{p}^{i,j} \in \mathcal{S}$, the extent to which $\mathbf{p}^{i,j} \approx \mathbf{p}^k$ for some $w_k \in \mathcal{E}$ depends on paraphrase error $\boldsymbol{\rho}^{k,\{i,j\}} = \mathbf{p}^k - \mathbf{p}^{i,j}$, that compares the induced distributions of $w_k$ and $\{w_i, w_j\}$. Thus the PMI vector difference $(\mathbf{p}^i + \mathbf{p}^j) - \mathbf{p}^k$ for any words $w_i, w_j, w_k \in \mathcal{E}$ comprises: $\boldsymbol{\varepsilon}^{ij}$ a component between $\mathbf{p}^i + \mathbf{p}^j$ and the surface $\mathcal{S}$ (reflecting word dependence); and $\boldsymbol{\rho}^{k,\{i,j\}}$ a component *along* the surface (reflecting paraphrase error). The latter captures a semantic relationship with $w_k$, which the former may obscure, irrespective of $w_k$. (Further geometric and probabilistic implications are considered in Appendix C.)

### 4.4 Linear combinations of PMI vectors find analogies

PMI vectors of analogy relationships "$w_a$ is to $w_{a^*}$ as $w_b$ is to $w_{b^*}$" have been proven [2] to satisfy:

$$\mathbf{p}^{b^*} \approx \mathbf{p}^{a^*} - \mathbf{p}^a + \mathbf{p}^b. \quad (9)$$

The proof builds on the concept of paraphrasing (with error terms similar to those in Section 4.3), comparing PMI vectors of analogous word pairs to show that $\mathbf{p}^a + \mathbf{p}^{b^*} \approx \mathbf{p}^{a^*} + \mathbf{p}^b$ and thus (9).

## 5 Encoding PMI: from PMI vectors to word embeddings

Understanding how high dimensional PMI vectors encode semantic properties desirable in word embeddings, we consider how they can be transferred to low dimensional representations. A key observation is that all PMI vector interactions, for similarity (7), paraphrases (8) and analogies (9), are *additive*, and are therefore preserved under *linear* projection. By comparison, the loss function of W2V (1) projects PMI vectors non-linearly, and that of GloVe (3) does project linearly, but not (necessarily) PMI vectors. Linear projection can be achieved by the least squares loss function:[3]

$$\ell_{LSQ} = \tfrac{1}{2} \sum_{i=1}^{n} \sum_{j=1}^{n} \left( \mathbf{w}_i^\top \mathbf{c}_j - \mathrm{PMI}(w_i, c_j) \right)^2. \quad (10)$$

$\ell_{LSQ}$ is minimised when $\nabla_{\mathbf{W}^\top} \ell_{LSQ} = (\mathbf{W}^\top \mathbf{C} - \mathbf{P})\mathbf{C}^\top = 0$, or $\mathbf{W}^\top = \mathbf{P}\,\mathbf{C}^\dagger$, for $\mathbf{C}^\dagger = \mathbf{C}^\top(\mathbf{C}\mathbf{C}^\top)^{-1}$ the *Moore–Penrose pseudoinverse* of $\mathbf{C}$. This explicit linear projection allows interactions performed between word embeddings, e.g. dot product, to be mapped to interactions between PMI vectors, and thereby semantically interpreted. However, we do better still by considering how $\mathbf{W}$ and $\mathbf{C}$ relate.

### 5.1 The relationship between W and C

Whilst W2V and GloVe train two embedding matrices, typically only $\mathbf{W}$ is used and $\mathbf{C}$ discarded. Thus, although relationships are learned between $\mathbf{W}$ and $\mathbf{C}$, they are tested between $\mathbf{W}$ and $\mathbf{W}$. If

$\mathbf{W}$ and $\mathbf{C}$ are equal, the distinction falls away, but that is not found to be the case in practice. Here, we consider why typically $\mathbf{W} \neq \mathbf{C}$ and, as such, what relationship between $\mathbf{W}$ and $\mathbf{C}$ does exist.

If the symmetric PMI matrix $\mathbf{P}$ is positive semi-definite (PSD), its closest low-rank approximation (minimising $\ell_{LSQ}$) is given by the eigendecomposition $\mathbf{P} = \mathbf{\Pi}\mathbf{\Lambda}\mathbf{\Pi}^\top$, $\mathbf{\Pi}, \mathbf{\Lambda} \in \mathbb{R}^{n \times n}$, $\mathbf{\Pi}^\top \mathbf{\Pi} = \mathbf{I}$; and $\ell_{LSQ}$ is minimised by $\mathbf{W} = \mathbf{C} = \mathbf{S}^{1/2}\mathbf{U}^\top$, where $\mathbf{S} \in \mathbb{R}^{d \times d}$, $\mathbf{U} \in \mathbb{R}^{d \times n}$ are $\mathbf{\Lambda}, \mathbf{\Pi}$, respectively, truncated to their $d$ largest eigenvalue components. Any matrix pair $\mathbf{W}^* = \mathbf{M}^\top \mathbf{W}$, $\mathbf{C}^* = \mathbf{M}^{-1}\mathbf{W}$, also minimises $\ell_{LSQ}$ (for any invertible $\mathbf{M} \in \mathbb{R}^{d \times d}$), but of these $\mathbf{W}, \mathbf{C}$ are unique (up to rotation and permutation) in satisfying $\mathbf{W} = \mathbf{C}$, a preferred solution for learning word embeddings since the number of free parameters is halved and consideration of whether to use $\mathbf{W}, \mathbf{C}$ or both falls away.

However, $\mathbf{P}$ is not typically PSD in practice and this preferred (real) factorisation does not exist since $\mathbf{P}$ has negative eigenvalues, $\mathbf{S}^{1/2}$ is complex and any $\mathbf{W}, \mathbf{C}$ minimising $\ell_{LSQ}$ with $\mathbf{W} = \mathbf{C}$ must also be complex. (Complex word embeddings arise elsewhere, e.g. [16, 22], but since the word embeddings we examine are real we keep to the real domain.) By implication, any $\mathbf{W}, \mathbf{C} \in \mathbb{R}^{d \times n}$ that minimise $\ell_{LSQ}$ *cannot be equal*, contradicting the assumption $\mathbf{W} = \mathbf{C}$ sometimes made [14, 4]. Returning to the eigendecomposition, if $\mathbf{S}$ contains the $d$ largest *absolute* eigenvalues and $\mathbf{U}$ the corresponding eigenvectors of $\mathbf{P}$, we define $\mathbf{I}' = sign(\mathbf{S})$ (i.e. $\mathbf{I}'_{ii} = \pm 1$) such that $\mathbf{S} = |\mathbf{S}|\mathbf{I}'$. Thus, $\mathbf{W} = |\mathbf{S}|^{1/2}\mathbf{U}^\top$ and $\mathbf{C} = \mathbf{I}'\mathbf{W}$ can be seen to minimise $\ell_{LSQ}$ (i.e. $\mathbf{W}^\top \mathbf{C} \approx \mathbf{P}$) with $\mathbf{W} \neq \mathbf{C}$ but where corresponding *rows* of $\mathbf{W}, \mathbf{C}$ (denoted by superscript) satisfy $\mathbf{W}^i = \pm \mathbf{C}^i$ (recall word embeddings $\mathbf{w}_i, \mathbf{c}_i$ are columns of $\mathbf{W}, \mathbf{C}$). Such $\mathbf{W}, \mathbf{C}$ can be seen as *quasi*-complex conjugate. Again, $\mathbf{W}, \mathbf{C}$ can be used to define a family of matrix pairs that minimise $\ell_{LSQ}$, of which $\mathbf{W}, \mathbf{C}$ themselves are a most parameter efficient choice, with $(n+1)d$ free parameters compared to $2nd$.

## 5.2 Interpreting embedding interactions

Various word embedding interactions are used to predict semantic relationships, e.g. cosine similarity [26] and 3CosMult [19], although typically with little theoretical justification. With a semantic understanding of PMI vector interactions (S.4) and the derived relationship $\mathbf{C} = \mathbf{I}'\mathbf{W}$, we now interpret commonly used word embedding interactions and evaluate the effect of combining embeddings of $\mathbf{W}$ only (e.g. $\mathbf{w}_i^\top \mathbf{w}_j$), rather than $\mathbf{W}$ and $\mathbf{C}$ (e.g. $\mathbf{w}_i^\top \mathbf{c}_j$). For use below, we note that $\mathbf{W}^\top \mathbf{C} = \mathbf{U}\mathbf{S}\mathbf{U}^\top$, $\mathbf{C}^\dagger = \mathbf{U}|\mathbf{S}|^{-1/2}\mathbf{I}'$ and define: *reconstruction error matrix* $\mathbf{E} = \mathbf{P} - \mathbf{W}^\top \mathbf{C}$, i.e. $\mathbf{E} = \overline{\mathbf{U}}\overline{\mathbf{S}}\overline{\mathbf{U}}^\top$ where $\overline{\mathbf{U}}, \overline{\mathbf{S}}$ contain the $n-d$ smallest absolute eigenvalue components of $\mathbf{\Pi}, \mathbf{\Sigma}$ (as omitted from $\mathbf{U}, \mathbf{S}$); $\mathbf{F} = \mathbf{U}(\frac{\mathbf{S} - |\mathbf{S}|}{2})\mathbf{U}^\top$, comprising the negative eigenvalue components of $\mathbf{P}$; and *mean embeddings* $\mathbf{a}_i$ as the columns of $\mathbf{A} = \frac{\mathbf{W} + \mathbf{C}}{2} = \mathbf{U}|\mathbf{S}|^{1/2}\mathbf{I}'' \in \mathbb{R}^{d \times n}$, where $\mathbf{I}'' = \frac{\mathbf{I} + \mathbf{I}'}{2}$ (i.e. $\mathbf{I}''_{ii} \in \{0,1\}$).

**Dot Product:** We compare the following interactions, associated with predicting relatedness:

$$
\begin{aligned}
\mathbf{W}, \mathbf{C} \ : \quad \mathbf{w}_i^\top \mathbf{c}_j &= \mathbf{U}^i \, \mathbf{S} \, \mathbf{U}^{j\top} & &= \mathbf{P}_{i,j} - \mathbf{E}_{i,j} \\
\mathbf{W}, \mathbf{W} \ : \quad \mathbf{w}_i^\top \mathbf{w}_j &= \mathbf{U}^i \, |\mathbf{S}| \, \mathbf{U}^{j\top} \ = \ \mathbf{U}^i(\mathbf{S} - (\mathbf{S} - |\mathbf{S}|))\mathbf{U}^{j\top} &= \mathbf{P}_{i,j} - \mathbf{E}_{i,j} - 2\,\mathbf{F}_{i,j} \\
\mathbf{A}, \mathbf{A} \ : \quad \mathbf{a}_i^\top \mathbf{a}_j &= \mathbf{U}^i \, |\mathbf{S}| \, \mathbf{I}'' \, \mathbf{U}^{j\top} \ = \ \mathbf{U}^i(\mathbf{S} - (\frac{\mathbf{S} - |\mathbf{S}|}{2}))\mathbf{U}^{j\top} &= \mathbf{P}_{i,j} - \mathbf{E}_{i,j} - \ \ \mathbf{F}_{i,j}
\end{aligned}
$$

This shows that $\mathbf{w}_i^\top \mathbf{w}_j$ *overestimates* the PMI approximation given by $\mathbf{w}_i^\top \mathbf{c}_j$ by twice any component relating to negative eigenvalues – an overestimation that is halved using mean embeddings, $\mathbf{a}_i^\top \mathbf{a}_j$.

**Difference sum:** $(\mathbf{w}_i - \mathbf{w}_j)^\top \mathbf{1} = (\mathbf{p}^i - \mathbf{p}^j)\mathbf{C}^\dagger \mathbf{1} = \sum_{k=1}^n \mathbf{x}_k \log \frac{p(c_k|w_i)}{p(c_k|w_j)}$, $\mathbf{x} = \mathbf{U}|\mathbf{S}|^{-1/2}\mathbf{I}'\mathbf{1}$

Thus, summing over the difference of embedding components compares to a KL divergence between induced distributions (and so *similarity*) more so than for PMI vectors (S.4.2) as dimensions are weighted by $\mathbf{x}_k$. However, unlike a KL divergence, $\mathbf{x}$ is not a probability distribution and does not vary with $\mathbf{w}_i$ or $\mathbf{w}_j$. We speculate that between $\mathbf{x}$ and the omitted probability weighting of the loss function, the dimensions of low probability words are down-weighted, mitigating the effect of "outliers" to which PMI is known to be sensitive [37], and loosely reflecting a KL divergence.

**Euclidean distance:** $\|\mathbf{w}_i - \mathbf{w}_j\|_2 = \|(\log \frac{p(\mathcal{E}|w_i)}{p(\mathcal{E}|w_j)})\mathbf{C}^\dagger\|_2$ shows no obvious meaning.

**Cosine similarity:** Surprisingly, $\frac{\mathbf{w}_i^\top \mathbf{w}_j}{\|\mathbf{w}_i\|\|\mathbf{w}_j\|}$, as often used effectively to predict word relatedness and/or similarity [33, 5], has no immediate semantic interpretation. However, recent work [3] proposes a more holistic description of relatedness than $\text{PMI}(w_i, w_j) > 0$ (S.4.1): that related words

Table 1: Accuracy in semantic tasks using different loss functions on the text8 corpus [24].

| Model | Loss | Relationship | Relatedness [1] | Similarity [1] | Analogy [25] |
|---|---|---|---|---|---|
| $W2V$ | W2V | $\mathbf{W}^\top \mathbf{C} \approx \mathbf{P}$ | .628 | .703 | .283 |
| $W{=}C$ | LSQ | $\mathbf{W}^\top \mathbf{W} \approx \mathbf{P}$ | .721 | .786 | .411 |
| $LSQ$ | LSQ | $\mathbf{W}^\top \mathbf{C} \approx \mathbf{P}$ | **.727** | **.791** | **.425** |

$(w_i, w_j)$ have multiple positive PMI vector components in common, because all words associated with any common semantic "theme" are also more likely to co-occur. The *strength* of relatedness (*similarity* being the extreme case) is given by the number of common word associations, as reflected in the dimensionality of a common aggregate PMI vector component, which projects to a common embedding component. The *magnitude* of such common component is not directly meaningful, but as relatedness increases and $w_i, w_j$ share more common word associations, the *angle* between their PMI vectors, and so too their embeddings, narrows, justifying the widespread use of cosine similarity.

Other statistical word embedding relationships assumed in [4] are considered in Appendix D.

# 6 Empirical evidence

Word embeddings (especially those of W2V) have been well empirically studied, with many experimental findings. Here we draw on previous results and run test experiments to provide empirical support for our main theoretical results:

1. Analogies form as linear relationships between linear projections of PMI vectors (S.4.4)

   Whilst previously explained in [2], we emphasise that their rationale for this well known phenomenon fits precisely within our broader explanation of W2V and GloVe embeddings. Further, re-ordering paraphrase questions is observed to materially affect prediction accuracy [23], which can be justified from the explanation provided in [2] (see Appendix E).

2. The linear projection of additive PMI vectors captures semantic properties more accurately than the non-linear projection of W2V (S.5).

   Several works consider alternatives to the W2V loss function [20, 21], but none isolates the effect of an equivalent linear loss function, which we therefore implement (detail below). Comparing models $W2V$ and $LSQ$ (Table 1) shows a material improvement across all semantic tasks from linear projection.

3. Word embedding matrices $\mathbf{W}$ and $\mathbf{C}$ are dissimilar (S.5.1).

   $\mathbf{W}$, $\mathbf{C}$ are typically found to differ, e.g. [26, 29, 28]. To demonstrate the difference, we include an experiment tying $\mathbf{W} = \mathbf{C}$. Comparing models $W{=}C$ and $LSQ$ (Table 1) shows a small but consistent improvement in the former despite a lower data-to-parameter ratio.

4. Dot products recover PMI with decreasing accuracy: $\mathbf{w}_i^\top \mathbf{c}_j \geq \mathbf{a}_i^\top \mathbf{a}_j \geq \mathbf{w}_i^\top \mathbf{w}_j$ (S.5.2).

   The use of average embeddings $\mathbf{a}_i^\top \mathbf{a}_j$ over $\mathbf{w}_i^\top \mathbf{w}_j$ is a well-known heuristic [29, 21]. More recently, [5] show that relatedness correlates noticeably better to $\mathbf{w}_i^\top \mathbf{c}_j$ than either of the "symmetric" choices ($\mathbf{a}_i^\top \mathbf{a}_j$ or $\mathbf{w}_i^\top \mathbf{w}_j$).

5. Relatedness is reflected by interactions between $\mathbf{W}$ and $\mathbf{C}$ embeddings, and similarity is reflected by interactions between $\mathbf{W}$ and $\mathbf{W}$. (S.5.2)

   Asr et al. [5] compare human judgements of similarity and relatedness to cosine similarity between combinations of $\mathbf{W}$, $\mathbf{C}$ and $\mathbf{A}$. The authors find a "very consistent" support for their conclusion that "WC ... best measures ... relatedness" and "similarity [is] best predicted by ... WW". An example is given for *house*: $\mathbf{w}_i^\top \mathbf{w}_j$ gives *mansion*, *farmhouse* and *cottage*, i.e. similar or synonymous words; $\mathbf{w}_i^\top \mathbf{c}_j$ gives *barn*, *residence*, *estate*, *kitchen*, i.e. related words.

**Models:** As we perform a standard comparison of loss functions, similar to [20, 21], we leave experimental details to Appendix F. In summary, we learn 500 dimensional embeddings from word co-occurrences extracted from a standard corpus ("text8" [24]). We implement loss function (1) explicitly as model $W2V$. Models $W{=}C$ and $LSQ$ use least squares loss (10), with constraint $\mathbf{W} = \mathbf{C}$ in the latter (see point 3 above). Evaluation on popular data sets [1, 25] uses the Gensim toolkit [32].

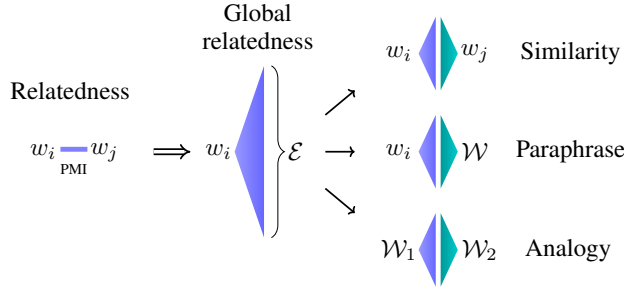

Figure 2: Interconnection between semantic relationships: relatedness is a base pairwise comparison (measured by PMI); *global relatedness* considers relatedness to all words (PMI vector); similarity, paraphrase and analogy depend on global relatedness between words ($w \in \mathcal{E}$) and word sets ($\mathcal{W} \subseteq \mathcal{E}$).

# 7 Discussion

Having established mathematical formulations for relatedness, similarity, paraphrase and analogy that explain how they are captured in word embeddings derived from PMI vectors (S.4), it can be seen that they also imply an interesting, hierarchical interplay between the semantic relationships themselves (Fig 2). At the core is *relatedness*, which correlates with PMI, both empirically [36, 6, 15] and intuitively (S.4.2). As a pairwise comparison of words, relatedness acts somewhat akin to a *kernel* (an actual kernel requires $\mathbf{P}$ to be PSD), allowing words to be considered numerically in terms of their relatedness to all words, as captured in a PMI vector, and compared according to how they each relate to all other words, or *globally relate*. Given this meta-comparison, we see that one word is *similar* to another if they are globally related (1-1); a *paraphrase* requires one word to globally relate to the joint occurrence of a set of words (1-$n$); and analogies arise when joint occurrences of word pairs are globally related ($n$-$n$). Continuing the "kernel" analogy, the PMI matrix mirrors a kernel matrix, and word embeddings the representations derived from *kernelised PCA* [34].

# 8 Conclusion

In this work, we take two previous results – the well known link between W2V embeddings and PMI [20], and a recent connection between PMI and analogies [2] – to show how the semantic properties of relatedness, similarity, paraphrase and analogy are captured in word embeddings that are linear projections of PMI vectors. The loss functions of W2V (2) and GloVe (3) approximate such a projection: non-linearly in the case of W2V and linearly projecting a variant of PMI in GloVe; explaining why their embeddings exhibit semantic properties useful in downstream tasks.

We derive a relationship between embedding matrices $\mathbf{W}$ and $\mathbf{C}$, enabling word embedding interactions (e.g. dot product) to be semantically interpreted and justifying the familiar *cosine similarity* as a measure of relatedness and similarity. Our theoretical results explain several empirical observations, e.g. why $\mathbf{W}$ and $\mathbf{C}$ are not found to be equal despite representing the same words, their symmetric treatment in the loss function and a symmetric PMI matrix; why mean embeddings ($\mathbf{A}$) are often found to outperform those from either $\mathbf{W}$ or $\mathbf{C}$; and why relatedness corresponds to interactions between $\mathbf{W}$ and $\mathbf{C}$, and similarity to interactions between $\mathbf{W}$ and $\mathbf{W}$.

We discover an interesting hierarchical structure between semantic relationships: with *relatedness* as a basic pairwise comparison, *similarity*, *paraphrase* and *analogy* are defined according to how target words each relate to all words. Error terms arise in the latter higher order relationships due to statistical dependence between words. Such errors can be interpreted geometrically with respect to the hypersurface $\mathcal{S}$ on which all PMI vectors lie, and can, in principle, be evaluated from higher order statistics (e.g trigrams co-occurrences).

Several further details of W2V and GloVe remain to be explained that we hope to address in future work, e.g. the weighting of PMI components over the context window [31], the exponent $3/4$ often applied to unigram distributions [26], the probability weighting in the loss function (S.5), and an interpretation of the weight vector $\mathbf{x}$ in embedding differences (S.5.2).

**Acknowledgements**

We thank Ivan Titov, Jonathan Mallinson and the anonymous reviewers for helpful comments. Carl Allen and Ivana Balažević were supported by the Centre for Doctoral Training in Data Science, funded by EPSRC (grant EP/L016427/1) and the University of Edinburgh.

## Footnotes

[1]We refer exclusively, throughout, to the more common implementation *Skipgram* with negative sampling.

[2]Whilst $w_i$, $w_j$ are *both* target words, by symmetry we can interchange roles of context and target words to compute $p(\mathcal{E}|w, w')$ based on the distribution of target words for which $w_i$ and $w_j$ are both context words.

[3]We note that the W2V and GloVe loss functions include probability weightings (as considered in [35]), which we omit for simplicity.

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
