[Supplementary Material]

# A  The W2V shift

The number of negative samples per observed word pair arises in the optimum of the W2V loss function (4) as the so-called *shift* term, $-\log k$. The shift is of a comparable magnitude to empirical PMI values [2] and causes dot product interactions to take more negative values, distorting embeddings relative to there being no shift term.

Under certain word embedding interactions, e.g. the linear combination associated with analogies, the shift terms cancel and thus have no effect [2]. However, elsewhere the shift term has been seen to have a detrimental impact on downstream task performance that removing it corrects [28].

Stemming from an arbitrarily chosen hyper-parameter $k$, the shift term is an artefact of the W2V algorithm that vanishes only if $k=1$. Setting that explicitly reduces the number of negative samples and results in poorer performance of the embeddings. Alternatively, $k$ can be *effectively* set to 1 by averaging the loss function components of each set of $k$ negative samples, i.e. multiplying by $\frac{1}{k}$.

# B  Properties of the PMI surface: proofs (Sec 4.1)

P1 **$\mathcal{S}$, and any subsurface of $\mathcal{S}$, is non-linear.**  This follows directly from the construction of $\mathcal{S}$, in particualr the application of the natural logarithm to the linear surface $\mathcal{R}$.

P2 **$\mathcal{S}$ contains the origin**  Follows from construction: $\mathbf{p} = \mathbf{p} \in \mathcal{Q}$ implies $\mathbf{1} = \frac{\mathbf{p}}{p(\mathcal{E})} \in \mathcal{R}$, and therefore $\mathbf{0} = \log \mathbf{1} \in \mathcal{S}$

P3 **Probability vector $\mathbf{q} \in \mathcal{Q}$ is normal to the tangent plane of $\mathcal{S}$**  at $\mathbf{s} = \log \frac{\mathbf{q}}{\mathbf{p}} \in \mathcal{S}$. Consider $\mathbf{q} = (q_1, ..., q_n) \in \mathcal{Q}$ as having free parameters $q_{j<n}$ that determine $q_n$, and let $\mathbf{J} \in \mathbb{R}^{n \times (n-1)}$ define the tangent plane to $\mathcal{S}$ at $\mathbf{s}$, i.e. $\mathbf{J}_{i,j} = \frac{\partial s_i}{\partial q_j}$. It can be seen that for $i < n$, $\mathbf{J}_{i,j} = q_j^{-1}$ if $i=j$, and $\mathbf{J}_{i,j} = 0$ otherwise; and that $\mathbf{J}_{n,j} = -(\sum_{j=1}^{n-1} q_j)^{-1} \forall j$. It follows that $\mathbf{q}^\top \mathbf{J} = \mathbf{0}$ and $\mathbf{q}$ is therefore normal to the tangent plane.

P4 **$\mathcal{S}$ does not intersect with the fully positive or fully negative orthants**  (excluding $\mathbf{0}$). This follows from the fact that components of one probability distribution, e.g. $p(\mathcal{E}|w_i)$, cannot *all* be greater (or *all* less) than their counterpart in another, e.g. $p(\mathcal{E})$. Any point in the fully positive or fully negative orthants would contradict this.

P5 **The sum of 2 points $\mathbf{s} + \mathbf{s}'$ lies in $\mathcal{S}$ only for certain $\mathbf{s}, \mathbf{s}' \in \mathcal{S}$.**  For probability vectors $\mathbf{p}$, $\mathbf{q}$, $\mathbf{q}' \in \mathcal{Q}$ and $\mathbf{s} = \log(\mathbf{q}/\mathbf{p})$, $\mathbf{s}' = \log(\mathbf{q}'/\mathbf{p}) \in \mathcal{S}$, we consider operations element-wise with correspondign vector elements denoted by lower case: $\mathbf{s} + \mathbf{s}' \in \mathcal{S}$ *iff* $s + s' = \log(q^*/p)$ for some probability vector $\mathbf{q}^* \in \mathcal{Q}$. Thus, $(q/p)(q'/p) = q^*/p$, or simply $(q/p)q' = q^*$, whereby components $(q/p)q'$ must sum to 1, or in vector terms $(\mathbf{q}/\mathbf{p})^\top \mathbf{q}' = 1$. since $\mathbf{q}'$ is a probability we can also say $(\mathbf{q}/\mathbf{p} - \mathbf{1})^\top \mathbf{q}' = 0$, and we have that $\mathbf{s} + \mathbf{s}' \in \mathcal{S}$ only if $\mathbf{s}' = \log(\mathbf{q}'/\mathbf{p}) \in \mathcal{S}$ with $\mathbf{q}'$ a probability vector orthogonal to $(\mathbf{q}/\mathbf{p}) - \mathbf{1}$. We see that the intersection of the hyperplane orthogonal to $(\mathbf{q}/\mathbf{p}) - \mathbf{1}$ and the simplex defines points $\mathbf{q}'$ that correspond to points in $\mathbf{s}' \in \mathcal{S}$ that can be added to $\mathbf{s}$, i.e. $\mathcal{S}_s$ (See Figs 3a and 3b). Trivially $\mathbf{0} \in \mathcal{S}_s$, $\forall \mathbf{s} \in \mathcal{S}$.

(a) Given point $\mathbf{s}' = \log \mathbf{q}'/\mathbf{p} \in \mathcal{S}$, those $\mathbf{q}'$ on simplex $\mathcal{Q}$ such that $\mathbf{s}' = \log \mathbf{q}'/\mathbf{p}$ satisfies $\mathbf{s} + \mathbf{s}' \in \mathcal{S}$.

(b) Subsurfaces $\mathcal{S}_s$ for a given point $\mathbf{s} \in \mathcal{S}$, and $\mathcal{S}_{s'}$ for any point $\mathbf{s}' \in \mathcal{S}_s$; showing also $\mathbf{s} + \mathbf{s}' \in \mathcal{S}$

Figure 3: Understanding the PMI surface $\mathcal{S}$.

## C Further Geometric properties of the PMI surface

Combining both geometric and probabilistic arguments shows:

1. PMI vectors of words $w_j$ that are both conditionally and marginally independent of word $w_i$, lie in a strict subsurface $\mathcal{S}_{\mathbf{p}^i} \subset \mathcal{S}$;

2. only $\mathbf{p}^j \in \mathcal{S}_{\mathbf{p}^i}$ add to $\mathbf{p}^i$ to give another point on the surface, specifically $\mathbf{p}^i + \mathbf{p}^j = \mathbf{p}^{i,j}$ corresponding to the joint occurrence of $w_i$ *and* $w_j$;

3. for any $\mathbf{p}^j \notin \mathcal{S}_{\mathbf{p}^i}$, $\mathbf{p}^i + \mathbf{p}^j$ is off the surface, separated from $\mathbf{p}^{\{w, w'\}}$ by an error vector $\epsilon_{i,j}$, reflecting statistical dependence between $w_i$ and $w_j$.

4. By symmetry, $\mathbf{s}' \in \mathcal{S}_{\mathbf{s}}$ *iff* $\mathbf{s} \in \mathcal{S}_{\mathbf{s}'}$, thus subsurfaces occur in distinct pairs $(\mathcal{S}_{\mathbf{s}}, \mathcal{S}_{\mathbf{s}'})$ that partition all points in $\mathcal{S}$. Furthermore, for any pair of points $\mathbf{t} \in \mathcal{S}_{\mathbf{s}}, \mathbf{t}' \in \mathcal{S}_{\mathbf{s}'}$, their sum $\mathbf{t} + \mathbf{t}' \in \mathcal{S}$ and every $s \in \mathcal{S}$ is the sum of a unique such pair, which we deonte $\mathcal{S}_{\mathbf{s}} \oplus \mathcal{S}_{\mathbf{s}'} = \mathcal{S}$, analogous to the Cartesian product.

## D Comparison to embedding relationships of previous works

The following relationships between W2V embeddings and probabilities are assumed in [4]:

$$\mathbf{w}_i = \mathbf{c}_i, \quad \log p(w_i) \approx \tfrac{\|\mathbf{w}_i\|^2}{2d} - \log Z \quad \text{and} \quad \log p(w_i, c_j) \approx \tfrac{\|\mathbf{w}_i + \mathbf{w}_j\|^2}{2d} - 2\log Z,$$

By rearranging $\mathbf{w}_i^\top \mathbf{c}_j \approx \mathrm{PMI}(w_i, c_j)$, as is claimed to follow from those above, we prove (below):

$$\log p(w_i) \approx \tfrac{-\mathbf{w}_i^\top \mathbf{c}_i}{2} + \tfrac{\log p(w_i, c_i)}{2} \quad \text{and} \quad \log p(w_i, c_j) \approx \tfrac{-(\mathbf{w}_i - \mathbf{w}_j)^\top (\mathbf{c}_i - \mathbf{c}_j)}{2} + \tfrac{\log p(w_i, c_i) p(w_j, c_j)}{2}.$$

Having previously shown that $\mathbf{w}_i \neq \mathbf{c}_i$ (Sec 5.1), if we nevertheless assume that equality for the sake of comparison, it can be seen that the relationships above differ fundamentally, e.g. having opposite sign. Also, the assumed *constant Z* can be seen to vary arbitrarily with the extent to which each word co-occurs with itself.

### D.1 Proofs

Noting $p(w_i) = p(c_i)$, since the difference is only the role attributed to a word, shows:

$$\mathbf{w}_i^\top \mathbf{c}_j \approx \log \tfrac{p(w_i, c_j)}{p(w_i) p(c_j)} = \log p(w_i, c_j) - \log p(w_i) - \log p(w_j) \tag{11}$$

If $i = j$, i.e. target and context words are the same, it follows that:

$$\mathbf{w}_i^\top \mathbf{c}_i \approx \log p(w_i, c_i) - 2\log p(w_i)$$

$$\text{i.e.} \quad \log p(w_i) \approx \tfrac{-\mathbf{w}_i^\top \mathbf{c}_i}{2} + \tfrac{\log p(w_i, c_i)}{2} \tag{12}$$

In the general case:

$$
\begin{aligned}
(\mathbf{w}_i - \mathbf{w}_j)^\top (\mathbf{c}_i - \mathbf{c}_j) &= \mathbf{w}_i^\top \mathbf{c}_i - \mathbf{w}_j^\top \mathbf{c}_i - \mathbf{w}_i^\top \mathbf{c}_j + \mathbf{w}_j^\top \mathbf{c}_j \\
&\overset{*}{=} \mathbf{w}_i^\top \mathbf{c}_i + \mathbf{w}_j^\top \mathbf{c}_j, -2\mathbf{w}_i^\top \mathbf{c}_j \\
&\overset{(11,12)}{\approx} (\log p(w_i, c_i) - 2\log p(w_i)) + (\log p(w_j, c_j) - 2\log p(w_j)) \\
&\qquad - 2(\log p(w_i, c_j) - \log p(w_i) - \log p(w_j)) \\
&= \log p(w_i, c_i) + \log p(w_j, c_j) - 2\log p(w_i, c_j)
\end{aligned}
$$

$$\text{thus} \quad \log p(w_i, c_j) \approx \tfrac{-(\mathbf{w}_i - \mathbf{w}_j)^\top (\mathbf{c}_i - \mathbf{c}_j)}{2} + \tfrac{\log p(w_i, c_i) p(w_j, c_j)}{2}. \tag{13}$$

The step marked * relies on $\mathbf{w}_i^\top \mathbf{c}_j = \mathbf{w}_i^\top (\mathbf{I}' \mathbf{w}_j) = (\mathbf{w}_i^\top \mathbf{I}') \mathbf{w}_j = \mathbf{c}_i^\top \mathbf{w}_j = \mathbf{w}_j^\top \mathbf{c}_i$, which follows from $\mathbf{C} = \mathbf{I}' \mathbf{W}$.

# E   Why order matters in analogies

Here, we develop the explanation of [2] to interpret the finding of Linzen [23] that some words within a particular analogy are more accurately predicted than others (see their "Reverse (add)").

From [2], we see that for analogy "$w_a$ is to $w_{a^*}$ as $w_b$ is to $w_{b^*}$", a "total error" term arises in the relationship $\mathbf{p}^{b^*} + \mathbf{p}^a = \mathbf{p}^{a^*} + \mathbf{p}^b$ between PMI vectors, and thus also word embeddings, due to statistical interactions between word pairs $\{w_a, w_{b^*}\}$ and $\{w_b, w_{a^*}\}$. Thus if $w_{b^*}$ is considered "missing" and to be predicted to complete the analogy, the statistical independence with $w_a$ is relevant, whereas if $w_b$ is to be predicted, statistical independence with $w_{a^*}$ is relevant. One of these may happen to exhibit higher independence, thus introduces less error and so be "easier to predict".

Separately, PMI vectors are unevenly distributed due to the non-uniform Zipf distribution of words. As such, some PMI vectors may happen to lie in more "cluttered" regions than others, an effect that may be exacerbated when projected to the far fewer dimensions of word embeddings. Thus, for the same magnitude error terms, words whose PMI vectors lie in more cluttered regions may be "harder to predict" due to many potential false positives nearby.

These two reasons explain (more concretely that the intuition of [23]) why the same analogy might more accurately be solved by predicting $w_b$ rather than $w_{b^*}$, or vice versa.

# F   Experimental details

## F.1   Training

PMI values are pre-computed from the corpus similarly to [29], substituting $-1$ for missing PMI values. We use the *text8* data set [24] containing $c.17$m tokens and $c.0.5$m unique words (sourced from the English Wikipedia dump, 03/03/06). 5 random word pairs (negative samples) are generated for each true word co-occurrence (positive sample) according to unigram word distributions. Dimensionality is 500. Words appearing less than 5 times are filtered and down-sampling is applied (see [26]). All models converged within 100 epochs (full passes over the PMI matrix). Learning rates that worked well were selected for each model: 0.01 for the least squares models, 0.007 for the W2V loss function. Results are averaged over 3 random seeds.

## F.2   Testing

Embeddings are evaluated on relatedness, similarity and analogy tasks using *WordSim353* [11, 1]. Ranking is by cosine similarity and evaluation compares Spearman's correlation between rankings and human-assigned similarity scores. Analogies use Google's analogy data set [25] of $c.20$k semantic and syntactic analogy questions "$w_a$ is to $w_{a^*}$ as $w_b$ is to ..?". Out-of-vocabulary words are filtered as standard [21]. Accuracy is computed by comparing $\mathrm{argmin}_{w_{b^*}} \|\mathbf{w}_a - \mathbf{w}_{a^*} - \mathbf{w}_b + \mathbf{w}_{b^*}\|$ to the labelled answer.