[Reviews · NeurIPS 2019]

Reviewer 1



The paper makes some observations about the word2vec and glove algorithms, specifically in light of their connection to factorizing the PMI matrix of co-occurrences. However it's not clear how the contributions are relevant, beyond them being a list of observations. Some sections explain the empirical relevance of these contributions, but the arguments are often convoluted (see contributions section). I also believe the claim made in the paper that much is left unknown about word2vec/glove (specifically what is learned and why is it useful) is exaggerated and in fact a lot of subsequent work has answered these questions. In particular the success of pre-embedding, unsupervised distributional methods is not new at all (see, e.g. the overview paper by Turney et al, 2010) Specifics: - Some parts of the paper simply re-iterate previous work, such as the beginning of sections 4 and 5. - How was the LSQ method in Table implemented exactly? - Please clarify 4.2 and 4.3 including the use of terms similarity and paraphrasing

Reviewer 2



This paper provides a new view of word embeddings. This paper introduces the notion of global relatedness, which is constructed by PMIs between the specific word and other context words. This paper shows the global relatedness can capture various semantic similarity, by considering geometric and probabilistic aspects of such vectors and their domain. Moreover, this paper shows low dimensional word embeddings built by word2vec or Glove can be viewed as the linear projection from the global relatedness vectors. This paper is well-written and easy to follow. This theoretical contribution is novel and provides a new tool to understand why word embeddings can capture various semantics of words. The originality and quality of this paper would be above the threshold.

Reviewer 3



This paper's view is novel and relatively solid. It provides a perspective for understanding the semantic similarity in word embedding, by (1) showing via space geometry that different semantic compositionality can be captured by PMI vectors (2) the linear projection between the PMI vectors and word embedding can preserve properties in (1). To me, the best part of the paper is that the author makes an effort to give a systematic and mathematically well-formed analysis addressing the frequently mentioned but not fully understood semantic issues in word embedding. The paper also derives a new model with LSQ loss in section 5 which achieves better performance and thus justified the previous analysis to some extent. My biggest concern lies in the absence of the understanding of COSINE similarity. If I understand the paper correctly, in section 4 the discrepancy between two PMI vectors is measured by their abstraction (\rho in Eq. 8 and \epsilon in Eq. 9), which is close to Euclidian distance rather than cosine distance (two vectors may have cosine similarity of 1 but can be very far from each other from the "abstraction or Euclidian" perspective). However, the cosine similarity is the most popular measure in practice (not only wording embedding), and since the paper claims an additive preserved linear transformation between the PMI vector and the word embedding, the cosine-related properties should also exist in PMI space and the author should also give a fair discussion there. Especially in section 4.3, the entire analysis based on the surface S does not suggest any property related to cosine similarity and thus is a bit skeptical whether this is really the case in practice. Even the results in Table 1 use cosine similarity as the measurement (as shown in Appendix F). Minor issue: the writing style could be improved. Specifically, (1) some of the explanations and statements in Section 4 and 5 are tedious and not easy to follow (e.g. a bunch of short equations within sentences which could be simplified). (2) Notations. It would be clear if using a single uppercase letter to represent the matrix, rather than "PMI", "SPMI" which look like matrix multiplication.

[Author Response · NeurIPS 2019]

**Reviewer #1:** Our key contribution is to explain mathematically how word embeddings of Glove/W2V capture semantic properties of words. Whilst some aspects relate to previous empirical observations or hypotheses e.g. a connection between relatedness and PMI (as acknowledged), to our knowledge no previous work *theoretically*: (i) explains the semantic properties of W2V/Glove embeddings as following from their low-rank projections of PMI vectors (S4); (ii) explores geometric properties of the space of PMI vectors and thus of word embeddings (S4.1); (iii) proves that relatedness, similarity, paraphrase and analogy (as defined) each correspond to mathematical relationships of PMI vectors with defined error terms that, e.g., in part explain the variability observed in analogy relationships [e.g. 22, Rogers et al. (2017)] (S4.2-4.4); (iv) concludes that those semantic relationships are best preserved by linear projection, under which embedding interactions can be probabilistically interpreted (S5) and several well-known observations/heuristics explained (S6); and (vi) from their formulations, derive a mathematical connection between relatedness, similarity, paraphrase and analogy. Regarding the referenced "observations" and "specifics":

- In S5.1, we prove $\mathbf{W} \neq \mathbf{C}$ since the opposite has been assumed [3,13]. Our result explains why tying $\mathbf{W} = \mathbf{C}$ gives good but sub-optimal results (since most eigenvalues are positive) and enables that sub-optimality to be quantified.

- In S5.2, we explicitly define the inherent error of low-rank approximation and the additional error due to taking the dot-product of embeddings from $\mathbf{W}$, as is often the case. We also quantify the effect of average embeddings.

- In S4, *similarity* corresponds to the "attributional similarity" of [37] and *paraphrasing* is as defined in [11, 2]. As such, "$w_a$ is similar to $w_b$" is equivalent to "$w_a$ paraphrases $\mathcal{W} = \{w_b\}$". We will make this more clear.

- In S4.2, we show that *subtraction* of PMI vectors equates to un-weighted KL divergence, thus where a PMI vector difference is small, the KL divergence is small and words are similar. This mathematically proves that (and how) PMI-based word embeddings instantiate the hypothesis that "similar row vectors in the word-context matrix indicate similar word meanings" for a general word-context matrix and unspecified vector-similarity measure [37].

- S4.3 shows how *addition* of PMI vectors (for words in a set $\mathcal{W}$) corresponds to identifying a paraphrase of words in $\mathcal{W}$, subject to their mutual independence, from which we can geometrically interpret the difference between the sum of PMI vectors and the PMI vector of the paraphrase word.

- S4 (beginning) justifies the view of word embeddings as projections of PMI vectors by extending [19]; S5 (beginning) motivates the use of linear projection. We believe both of these are original perspectives for W2V/Glove.

- The *LSQ* model implements loss function (11), details are in S6 (end) and Appendix F. We will include that all implementations use PyTorch with the Adam optimiser and review wording to ensure it is comprehensive and clear.

Many works investigate W2V/Glove and their embeddings [18, 22, 4, 13, 3, 8, 17, 11, 2, 9], indicating that they do "remain poorly understood" [28]. We provide, *inter alia*, a first explicit mathematical understanding of what W2V/Glove embedding parameters represent and the semantic relationships they capture. We believe our work is relevant as these algorithms have been adopted by other domains [30, 31], their embeddings are ubiquitous and the presence of linear relationships between word embeddings has been recently questioned (Rogers et al. (2017), Schluter (2018)). We provide interpretability of W2V and Glove, enabling principled improvement of word embeddings (as shown with LSQ loss) and comparison metrics; and interpretability in down-stream tasks. We will make our contributions more clear.

**Reviewer #2:** *Global relatedness* (GR) refers to when two words (or word sets) induce similar distributions over all other words, which underpins the definitions of similarity, paraphrase and analogy. GR manifests geometrically as a small difference between one PMI vector (or sum of PMI vectors) and another, meaning that the associated KL divergence is small and the relevant semantic relationship (similarity, paraphrase or analogy) exists. Since the interactions of PMI vectors associated with those semantic relationships are linear, when PMI vectors are projected linearly to a lower dimension, those relationships are necessarily maintained and GR remains identifiable between the low-dimensional representations, i.e. word embeddings. In summary, as a kind of equivalence measure within semantic relationships, global relatedness corresponds to small vector difference in PMI space and thus also in the space of word embeddings given a sufficiently homomorphic (e.g. linear) projection. We will make this more clear in the paper.

**Reviewer #3:** Section 5.2 (end) considers the interpretation of cosine similarity. While cosine similarity is not found to have an obvious probabilistic interpretation, we conjecture (based on the other mathematical relationships derived) that it serves as a blended measure of similarity and relatedness. The dot product numerator approximates relatedness, but, as you say, can be high even if vectors are far apart in Euclidean distance (e.g. words that are related but not similar). By normalising, cosine similarity requires the angle between embeddings to be small, which is closer to a similarity test, thus cosine similarity appears to evaluate somewhere between relatedness and similarity, explaining why it has been used to measure both [33, 4]. To clarify, the difference between two PMI vectors is indeed given by $\rho$ of Eq. 8 (note: $\epsilon$ of Eq. 9 applies only when PMI vectors are added to find paraphrases). In PMI space, an appropriately weighted sum over $\rho$ components gives a KL divergence and thus a probabilistic measure of word similarity. We suggest (S5.2) that, by dropping low probability dimensions, the low-dimension projection to word embeddings approximates such a probabilistic weighting over dimensions. In future work we plan to investigate these interactions in finer detail.
"Minor issues": we will improve readability of sections 4 and 5 and simplify notation as suggested.

[Meta-Review · NeurIPS 2019]

I like this paper and I want to see it in the conference. It provides a valuable new perspective on the theoretical properties of embedding spaces and how they relate to word distributions. I interpret the low-scoring review as more of a lack of interest on the part of the reviewer rather than an indication of the quality of the contribution. There has been a substantial literature on theoretical approaches to vector embedding models since the Turney and Pantel 2010 paper, and I believe this work is a solid and valuable addition. R3's concern about cosine distance vs. Euclidean is real and should be addressed in some fashion, but is not a game changer. Personally, I would speculate that most vectors are of similar length except for a few extremely frequent words, so that cosine and Euclidean usually aren't that different.